# The role of fluctuations in determining cellular network thermodynamics

**Joseph B. Hubbard, Michael Halter, Swarnavo Sarkar, Anne L. Plant** [ID] *

Biosystems and Biomaterials Division, National Institute of Standards and Technology, Gaithersburg, MD, United States of America

* anne.plant@nist.gov

## Abstract

The steady state distributions of phenotypic responses within an isogenic population of cells result from both deterministic and stochastic characteristics of biochemical networks. A biochemical network can be characterized by a multidimensional potential landscape based on the distribution of responses and a diffusion matrix of the correlated dynamic fluctuations between N-numbers of intracellular network variables. In this work, we develop a thermodynamic description of biological networks at the level of microscopic interactions between network variables. The Boltzmann H-function defines the rate of free energy dissipation of a network system and provides a framework for determining the heat associated with the nonequilibrium steady state and its network components. The magnitudes of the landscape gradients and the dynamic correlated fluctuations of network variables are experimentally accessible. We describe the use of Fokker-Planck dynamics to calculate housekeeping heat from the experimental data by a method that we refer to as Thermo-FP. The method provides insight into the composition of the network and the relative thermodynamic contributions from network components. We surmise that these thermodynamic quantities allow determination of the relative importance of network components to overall network control. We conjecture that there is an upper limit to the rate of dissipative heat produced by a biological system that is associated with system size or modularity, and we show that the dissipative heat has a lower bound.

## Introduction

Measurements of individual cells within a population indicate that phenotypic differences between isogenic cells is common, even when they are in a homogeneous and stable environment. Despite differences between individuals, populations of cells can manifest apparently stable distributions of phenotypic expression. The measured phenotypic parameters can include concentration of protein products of gene expression, indicators associated with promoter activation, RNA transcripts, and complex cell traits such as morphology. The steady state distribution of phenotypes observed from single cell analysis is a probability density function and can be represented as a potential energy landscape [1–5]. Living cells are a clear example of a nonequilibrium system [6].

Studies that involve imaging of live cells reveal that despite the observed consistency of a steady state distribution of phenotypes across a population of cells, there can be significant

**Data Availability Statement:** No data are used in this theoretical study.

**Funding:** The authors received no specific funding for this work.

**Competing interests:** The authors have declared that no competing interests exist.

dynamic variability in each cell and from one cell to another [3, 7–13]. A number of studies provide direct evidence that the distribution of phenotypes in steady state distributions is ergodic in that subsequent to a transient perturbation the population will eventually relax to the steady state distribution [3, 14–18] when culture conditions are kept constant. The recapitulation of the steady state distribution, even after single cell cloning, demonstrates that each cell or its progeny can explore all microstates on the landscape. The population response appears invariant, but the individual entities (cells) that comprise the population present a dynamic, random expression of phenotypes, which ultimately results in the heterogeneity of the population. These are characteristics of many physical systems that can be well-described by statistical mechanics. The variations in populations of cells are often attributed to stochastic fluctuations, or noise, due to small numbers of molecules associated with transcription and translation. However, fluctuations in small numbers of molecules is not the only source of this variation [9, 19–21]; correlation analysis has suggested that the main source of noise may be upstream regulatory components [14]. Ensembles of biochemicals are responsible for producing an observed phenotype [9, 21, 22], and these network components demonstrate coordinated concentration response functions. Importantly, dynamic fluctuations of network components, and correlations in fluctuations among multiple network components, are defining features of regulated networks [2, 13, 16, 21, 23–31]. An abbreviated biochemical network system is depicted in the schematic in Fig 1, based on the interactions between transcription factors involved in maintaining pluripotency [32], and provides an illustration of some of the general network features that our model addresses.

Identification of network components and the nature of their interdependence remains a challenge. While some "omics" analysis methods can probe many variables simultaneously and in some cases at the level of individual cells, they provide only a snapshot in time. While these methods allow determination of the coincidence of molecular species, without dynamic data in individual cells that can reveal the rates and magnitudes of the interactions between network species, it is impossible to determine unambiguously the relationship between them or to determine a physical mechanistic basis for control of the network [11, 21, 33, 34]. A promising approach for observing single cell dynamics is imaging of live cells over time, which provides access to fluctuations in phenotypic expression in individual cells across a population [10–12, 16, 26]. Live cell imaging can also provide spatial and temporal information across scales and can provide ancillary information such as direct observation of the timing of cell division. We have previously used quantitative microscopy [3, 8] to track temporal responses of populations of individual cells, and Langevin/Fokker-Planck (L/FP) dynamics to analyze the population distributions as potential energy landscapes [3]. A cell line expressing green fluorescent protein (GFP) associated with the promoter for the gene for the extracellular matrix protein, tenascin C, was monitored by live cell fluorescence microscopy. The steady state distribution of expression levels of GFP in individual cells allowed construction of the potential landscape for the population. Changes in fluorescence intensity in individual cells were quantified at 15-minute intervals. By measuring real time trajectories of the activity of the gene promoter, we determined the diffusion coefficient for fluctuations in the promoter. We selected four subpopulations of cells by flow sorting, and by employing L/FP dynamics, accurately predicted the time-dependent relaxation of the subpopulations to the steady state distribution. The subpopulations were selected based on different levels of promoter activity and they relaxed to steady state with very different kinetics, but the diffusion coefficient provided excellent predictions of all relaxation kinetics with no adjustable parameters [3]. The study demonstrated that accurate experimental data on gene expression fluctuations can be collected by fluorescence microscopy, that data at the cellular level can provide additional details about control of gene expression that cannot be determined at the population level, and that FP

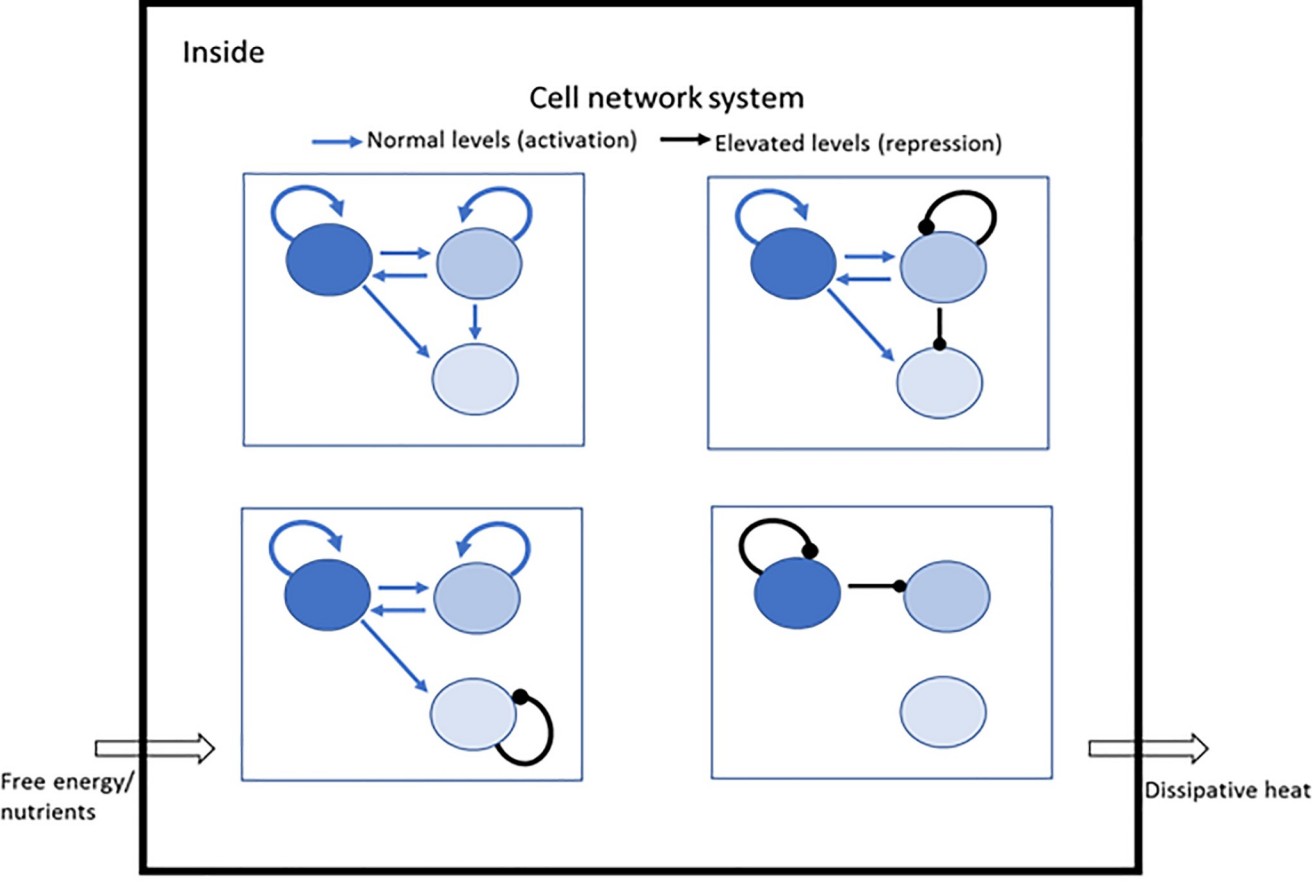

**Fig 1. Schematic of an intracellular network.** This is based on concepts presented by Rizzino and Wuebben [32] on control of pluripotency, and consists of N = 3 variable components, or dimensions, of a network with activation and repression relationships as indicated. As described in the text, these relationships define the measurable covariances in an NxN diffusion matrix, which contribute to the thermodynamics of the network together with the gradients of a landscape (which are derived from the multi-dimensional steady state probability density). The nonequilibrium steady state in this open thermodynamic system is supported by an influx of free energy from outside the system, which is dissipated as heat.

dynamics is an appropriate approach to probing intracellular control mechanisms from experimental observations.

While such a one-dimensional landscape is a simple manifestation of a complex adaptive system, it provides little insight into the mechanisms responsible for establishing and maintaining the stability and dynamics of the steady state distribution because of the contributions from unseen variables. In the current theoretical work, we consider a network of N numbers of variables. An N-dimensional statistical thermodynamics analysis allows fundamental questions about the thermodynamic controls in a coordinated network system to be addressed, including: how to identify the most important components of a regulated network, what is the relative thermodynamic contribution of different network components, and what is the thermodynamic price of homeostasis of a regulated network?

We present an experimentally accessible theoretical framework, which we refer to as Thermo-FP, that demonstrates that the rate of free energy dissipation associated with maintaining a nonequilibrium steady state network of intracellular reactions (i.e. the housekeeping heat) can be determined by the covariances in the temporal fluctuations in the components of the network together with the gradients of the potential landscape. This analysis is enabled by

a recent application of FP dynamics [35] to demonstrate that the Boltzmann H-function explicitly connects the relative entropy of a relaxing population with the rate of dissipation of free energy involved in maintaining the network at steady state.

## Results

### The theoretical framework

In this work, we develop a thermodynamic description of biological networks at the level of microscopic interactions between network variables. Experimentally accessible measurements of network variables at the level of single cells can provide data about the fluctuations in, and dynamic interactions between, those network variables. We build on the rigorous Kullback-Leibler based definition of relative free energy presented by Rao and Esposito [36]. Instead of using a master equation, we apply FP dynamics and the Boltzmann H-function to describe the rate of approach to steady state in terms of the dynamical fluctuating behavior of network components, and the thermodynamic quantities that can be derived from the correlations in fluctuations of the network variables. We call this approach for interpreting the kinetics of phenotype expression Thermo-FP. This coarse-grained approach diverges from a master equation approach, bypassing the need to have detailed knowledge of explicit reaction steps which are often difficult to know with confidence; it allows evaluation of the thermodynamics of complex networks for which there is insufficient knowledge to write chemical rate equations. This approach emphasizes the role of fluctuations or "noise" in controlling a biological system and provides a direct link between the dynamic correlations between network variables and a thermodynamic understanding of network size, composition, and relative importance of network variables as will be shown. This is possible through experimentally measurable quantities.

We present Thermo-FP to describe the evolution to, and maintenance of, a steady state of an $N$-dimensional network probability distribution with $N$ x $N$ diffusion matrices of variances and covariances in and between network variables. The magnitude of the coupled fluctuations between the different network variables, i.e., the covariances that comprise the diffusion matrix, is a measure of the strength of the physical and functional interaction between those variables [21]. The coordinated relationships between network components create an organized structure, thus reducing system entropy and resulting in the production of dissipative heat [37, 38].

Here we treat multidimensional landscapes of biochemical networks by applying the mathematical properties of positive definite quadratic forms and normal mode analysis common to mathematical physics and statistical mechanics [39]. A diffusion matrix of the dynamic covariances between the multiple variables of the network can be rotated through normal mode analysis to identify complex collective modes of network variables. We show that the product of each eigenvalue of the rotated diffusion matrix with the square of the gradient of the rotated multidimensional landscape allows determination of the contribution each degree of freedom makes to the rate of dissipation of heat that maintains the regulated network of reactants.

We begin by showing that the Boltzmann H function describes the free energy of a population relaxing by Thermo-FP dynamics to a steady state distribution.

### The steady state landscape and the Boltzmann H function

The free energy of a system approaching a nonequilibrium steady state can be identified with the Boltzmann H-function, $H(t)$, and $- dH/dt$ is the rate of dissipation of free energy as a system that is transiently perturbed away from its steady state relaxes to a steady state distribution, $W_{ss}$.

$H(t)$ is a relative free energy defined by

$$H(t, \tau) = k_B T \int d^N x W_1(\{x\}, t) \ln \frac{W_1(\{x\}, t)}{W_2(\{x\}, t + \tau)} \tag{1}$$

in terms of probability densities, where $W_1(\{x\},t)$ is the probability density of the microstates at some time, $t$ during relaxation, $W_2(\{x\},t+\tau)$ is the probability density at time $t+\tau$ (depicted schematically in Fig 2), and $T$ is the thermodynamic temperature of the network system in contact with an isothermal heat reservoir [40]. We assume that no entropy production is associated with temperature gradients. Thus, the relative free energy of the system is defined by the average of the logarithm of the ratio of the occupation probabilities of the microstates, $\{x\}$, of the distribution. Keeping in mind that time-dependent relaxation is determined by stochastic fluctuations of characteristic rates, a series of probability density functions is observed over time (Fig 1), each of which can be described by FP dynamics in terms of a potential force, or drift, term and a diffusive term at every microstate; the former corresponds to the gradient of the landscape on which the microstate resides, and the latter is described in an $N$ by $N$ diffusion matrix of variances and covariances of the fluctuations of the N network variables (see S1 Text). As the perturbed system relaxes, $H(t)$ decreases with time ($\frac{dH(t)}{dt} \leq 0$) and is a minimum at steady state, $SS$.

The time derivative of the Boltzmann H-function, $\frac{dH(t)}{dt}$ can be written as a quadratic form [35, 41], consistent with FP dynamics:

$$\frac{dH(t)}{dt} = -\frac{k_B T}{2} \int d^N x W(\{x\}, t) \sum_{i,j}^{N} \frac{\partial}{\partial x_i} \ln R \cdot D_{ij}(\{x\}) \cdot \frac{\partial}{\partial x_j} \ln R \tag{2}$$

where $\ln R = \ln \frac{W(\{x\},t)}{W_{SS}(\{x\})}$ is the microscopic relative free energy associated with the microstate {x}

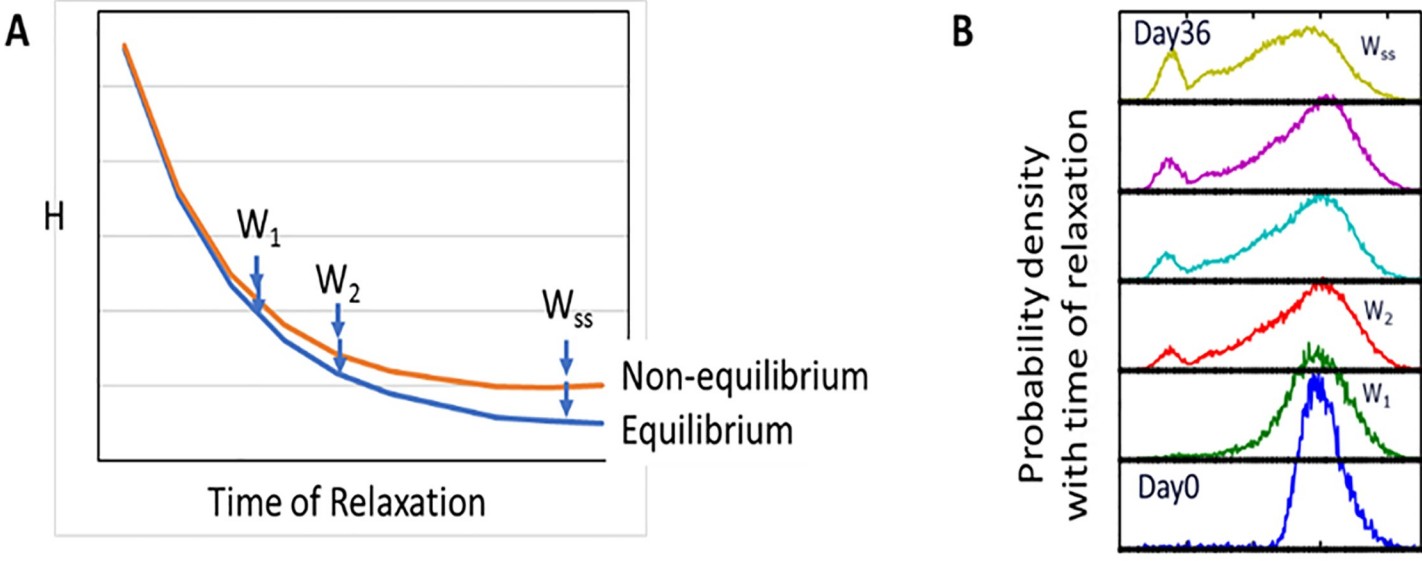

**Fig 2. A steady state distribution, if transiently perturbed, will relax back to the steady state distribution.** (Left) The rate of dissipation of free energy during relaxation, $\frac{dH(t)}{dt}$, approaches 0 as the steady state ($W_{ss}$) is reached. A thermodynamically open, nonequilibrium system is less entropic, and of higher free energy, than an equilibrium system. (Right) $H(t)$ corresponds to population distributions (shown here for 1-dimensional distributions). A subpopulation resulting from cell sorting (Day 0) is allowed to relax, and over time, the population achieves the initial steady state distribution from which the subpopulation was taken (3).

of the landscape. The quadratic form in the integrand of Eq 2 is the inner product of the probability current and the thermodynamic force, which explicitly defines the contribution to the rate of free energy dissipation due to fluctuations in network variables. The gradients of the landscape, $\frac{\partial}{\partial x_i} \ln R$ and $\frac{\partial}{\partial x_j} \ln R$ are analogous to the gradients of a chemical potential; and $D_{ij}$ is a diffusion matrix of experimentally determined dynamic covariances between the $N$ variables [41–44]. The $N$-dimensional diffusion matrix, $D_{ij}$, reflects the fluctuations in each of $N$ network components, i.e. the diagonal elements (variances), and the dynamic correlations (covariances) between the $N$ variables, i.e. the off-diagonal elements of the diffusion matrix.

## The energetic components of the non-equilibrium steady state

By setting $R(\{x\}, t) = \frac{W(\{x\}, t)}{W_{ss}(\{x\})}$ in Eq 2, where $W_{ss}(\{x\})$ refers to a time-invariant steady state, we see that Thermo-FP dynamics ensures that $\frac{dH(t)}{dt}$ vanishes as the stationary state is approached and $R$ approaches unity and $\ln R (\{x\}, t)$ approaches 0.

At equilibrium, the rate of increase in entropy and the rate of dissipation of free energy will approach zero. Living biological systems do not exist at equilibrium, and instead reach a non-equilibrium steady state with reduced entropy and higher free energy compared to an equilibrium state. The equilibrium state is defined by detailed balance, in that the probability flux between any pair of points on the landscape is equal to the probability flux in the reverse direction; in the nonequilibrium state there is a lack of reversibility. Regulated interactions between network components create a coordinated set of reactions; this allows the system to respond dynamically to arbitrary perturbations in order to recover homeostasis. This molecular organization reduces entropy and keeps the network system in a nonequilibrium steady state that is maintained by the injection of free energy from outside the network system; that energy is dissipated by the system as heat [37, 38]. During relaxation to an equilibrium steady state, $\frac{dH(t)}{dt}$ is the rate of free energy dissipated from the production of entropy during approach to the steady state, i.e., $dH/dt = -T\dot{Q}_{EN}$ where $\dot{Q}_{EN}$ is the entropy production rate. During relaxation to a nonequilibrium steady state distribution, $\frac{dH(t)}{dt}$ is the rate of free energy dissipated from two sources, namely $\dot{Q}_{EN}$, and the rate of dissipation of heat from the free energy that is required to maintain the nonequilibrium steady state, $\dot{Q}_{HK}$, the housekeeping heat. (The dot indicates the time rate of change of the quantity). Thus, for a nonequilibrium system: $\dot{Q}_{HK}(t) \geq 0$ [38], $\dot{Q}_{EN}(t) \geq 0$ [38], and $dH/dt \leq 0$ [44], and

$$\dot{Q}_{HK}(t) = T\dot{Q}_{EN}(t) + \frac{dH(t)}{dt} \tag{3}$$

Before reaching steady state:

$$\dot{Q}_{HK}(t) = \frac{k_B T}{2} \int d^N x W(\{x\}, t) \sum_{i,j}^{N} \left[ \frac{\partial}{\partial x_i} \ln \frac{W_{ss}(\{x\})}{W_{eq}(\{x\})} \cdot D_{ij}(\{x\}) \cdot \frac{\partial}{\partial x_j} \ln \frac{W(\{x\}, t)}{W_{eq}(\{x\})} \right] \geq 0 \quad (4)$$

where $\dot{Q}_{HK}(t)$ is the instantaneous housekeeping heat and is a functional that varies with the probability distribution, $W(\{x\}, t)$; $W_{ss}$ is the nonequilibrium steady state; and $W_{eq}$ is the equilibrium steady state (where detailed balance holds). The first gradient term in the bracket before the diffusion tensor is the thermodynamic force at steady state and the remainder of the term is the probability flux. The diffusion term, $D_{ij}(\{x\})$, explicitly connects network component fluctuations to housekeeping heat.

The instantaneous housekeeping heat $\dot{Q}_{HK}$ as shown in Eq 4 is a bilinear form, which at steady state reaches a constant value:

$$\dot{Q}_{HK} = \frac{k_B T}{2} \int d^N x W_{ss}(\{x\}) \sum_{i,j}^{N} \left[ \frac{\partial}{\partial x_i} \ln \frac{W_{ss}(\{x\})}{W_{eq}(\{x\})} \cdot D_{ij}(\{x\}) \cdot \frac{\partial}{\partial x_j} \ln \frac{W_{ss}(\{x\})}{W_{eq}(\{x\})} \right] \geq 0 \quad (5)$$

The steady state expression for $\dot{Q}_{HK}$ is a quadratic form for any trial steady state distribution. The steady state distribution that is a solution of the multidimensional FP equation for the cellular network minimizes the quadratic form in Eq 5.

The Boltzmann H function thus provides us with a measure of the housekeeping heat, $\dot{Q}_{HK}$ introduced by Oono and Paniconi [45] and further developed by others [38, 46]. The positive, semi-definite quadratic form of $\dot{Q}_{HK}$ at steady state is immediately apparent from Eqs 4 and 5.

## Normal mode representation of the diffusion matrix

The nonequilibrium steady state distribution of a cell population can be thought of as composed of microstates, $\{x\}$, i.e. the different ways the population can achieve its continuum of phenotypic expressions. The phenotypic expressions of the population are determined by the dynamic interactions of the components of the network; these interactions are represented in the N x N diffusion matrix, $D_{ij}(\{x\})$, as the autocorrelations and the dynamic covariances of the fluctuations of the activities of the N network variables. The dynamic covariances are the mean square fluctuations about the average relative expression levels divided by time. The H function provides us with the time-derivative of the relative free energy of the system, from which we can identify $\dot{Q}_{HK}(t)$, the rate at which heat is dissipated during the maintenance of the nonequilibrium steady state of the network. We will use these relationships to determine the relative rate of heat dissipation by the various network components to support the homeostatic steady state distribution.

The normal modes of the diffusion matrix result from rotation of the matrix $D_{ij}(\{x\})$ to a diagonal form and provide us with a matrix $D_{ii}(\{x^*\})$ in which the eigenvalues of all diagonal elements are greater than 0 and the off-diagonal elements are equal to 0, i.e., the components of the matrix are independent of one another. The purpose of this transformation is to identify combinations of network components as composite variables that are the major contributors to the rate of heat dissipation of the system; these are effectively the degrees of freedom. The original diffusion matrix consists of self- and cross-correlations between network variables, where some of the cross-correlations can be positive and some negative. Rotation of the matrix allows us to define composite variables (degrees of freedom) as clusters of $N$-wise interactions between the $N$ components of the network. The rotation operation guarantees eigenvalues that are positive as well as eigenvectors that are mutually orthogonal, and yields an expression for the energetics of the system at steady state as

$$\dot{Q}_{HK} = k_B T \int d^N x^* W_{SS}(\{x^*\}) \cdot \sum_i^N \lambda_i(\{x^*\}) V_i^2(\{x^*\}) \quad (6)$$

In Eq 6, $\lambda_i(\{x^*\})$ are the eigenvalues of the rotated diffusion matrix and represent interactions between network components and the magnitude of their coordinated fluctuations. Eigenvectors represents ways, i.e., modes, in which network components are organized with respect to their interactions with one another. The eigenvalues are the diffusion coefficients associated with those independent collective modes of the network components. The term $V_i(\{x^*\})$ is equal to the gradient of the potential defined by the rotated landscape and expressed

as $[V_i^2(\{x^*\})] = \left[\frac{\partial}{\partial x_i^*}\ln\frac{W_{ss}}{W_{eq}}(\{x^*\})\right]^2$. It should be noted that the eigenvalues as well as the gradients of the potentials are functions of the entire coordinate set of $x^*$. Eq 6 shows that the rate of free energy dissipation to maintain the nonequilibrium steady state will be largest when eigenvalues are large because of strong dynamic interactions between components of the network and when those interactions are occurring in a part of the landscape that is characterized by steep gradients. The rotated diffusion matrix produces positive eigenvalues and corresponding orthogonal eigenvectors, which simplifies the next step which is to sum the most important contributors to the rate of free energy utilization and heat dissipation.

## Network contributions to $\dot{Q}_{HK}$

$\dot{Q}_{HK}$ in Eq 6 is the rate at which the N-dimensional network is dissipating heat associated with the maintenance of the nonequilibrium steady state. It can also be regarded as the rate at which external free energy is injected into the network. The summation term on the right side of Eq 6 is a positive definite quadratic form; in keeping with statistical thermodynamics we use this term to determine the energy that each of the degrees of freedom contribute to the system. The magnitude of $\lambda_i(\{x^*\})$ and $[V_i(\{x^*\})]^2$ in $\dot{Q}_{HK}$ integrated over the probability distribution defines the rate of heat dissipation by the various components of the homeostatic network. This treatment implies that some interactions between network components or variables are more dissipative than others and therefore are more important thermodynamic contributors to the steady state. Below we will address the significance of this.

If we consider each composite network variable that is identified by rotation of the matrix as a contributor of a degree of freedom, their sum, $C$, represents the total rate of heat dissipation associated with the system as shown in Eq 7.

$$C = \sum_i^N \lambda_i [V_i^*]^2 \tag{7}$$

Each term is an implicit function of $x^*$ and reflects a heat that is dependent on the magnitudes of the eigenvalues of the matrix, $\lambda_i$, and the square of the gradient, $V_i^*$, of the landscape that corresponds to the rotated coordinate system. Together these terms constitute a quadratic representation of the microscopic dissipation of the regulated circuit, a sequence of partial sums that increases monotonically as the number of terms increases. We are proposing that reactions that involve high rates of free energy dissipation are more important to the function of the cell vis a vis stability and adaptability.

We now make an assumption that for a biological system, there will likely be an upper bound, $C_{UB}$, on the local rate of dissipative heat produced. It is well established that living systems are sensitive to nonoptimal temperatures [47], so the rate of heat production cannot exceed the rate at which heat can be dissipated without a failure of the system. This upper limit may arise from the temperature associated with heat generation or the limited rate of transport of energy or matter from the environment into the system. The value of the sum of the energetics of the network components thus cannot exceed the upper bound denoted by:

$$C_{UB} \geq C = \lambda_1 V_1^* 2 + \lambda_2 V_2^* 2 + \lambda_3 V_3^* 2 + \cdots \lambda_N V_N^* 2 \tag{8}$$

Each of the elements in Eq 8 is a contributor to the summation of network dissipative heat. This treatment thus provides us, in principle, with an experimentally tractable way of assessing the relative contribution that each multivariable component makes to the rate of heat dissipation in the maintenance of the network, and the free energy cost associated with keeping the

network in homeostasis. When the number of network dimensions, $N$, is sufficiently large, $C$ reaches a limit, $C_{UB}$ and after integrating over the multidimensional probability distribution this constant rate is identical to $\dot{Q}_{HK}$. A geometric approach for estimating the size of $N$ required for convergence of $C$ to $C_{UB}$ is described in S1 Text.

The upper limit to the dissipative heat $C_{UB}$ can constrain the maximum dimension of components in a cellular network. However, at each microstate {x} of the network all the $N$ eigenvalues would not be strictly greater than zero. In fact, if a cellular network is modular then only a small subset of the $N$ network components will have non-zero eigenvalues at each microstate. There is strong evidence that biological networks are hierarchical and modular in their topology [48], which allows for a large number of network components without crossing the dissipation rate threshold.

In addition to the assumption of an upper bound on dissipative heat, we utilize a generalization of the maximum entropy principle to show that the homeostatic heat generation rate, $\dot{Q}_{HK}$, also has a lower bound [49], i.e., there exists a finite dissipative gap between the non-equilibrium stationary state and the equilibrium (detailed balance) distribution (see S1 Text).

## Discussion

It has been frequently noted that stochastic fluctuations in molecular components in individual cells are important to regulatory mechanisms in one-dimensional systems [14, 26, 50, 51] and in multidimensional networks [23, 27, 29–31, 52, 53]. For example, Mojtahedi et. al. [31] analyzed transitions in lineage progression in a population that was attributed to a fluctuation-driven disappearance of an attractor basin. But to our knowledge, the current work is the first to show the direct relationship between fluctuations in and dynamic covariances between network variables and the thermodynamic quantities that contribute to housekeeping heat.

We have limited our theoretical Thermo-FP approach to one that is experimentally tractable. We have shown here how the use of a potential landscape and a diffusion matrix provides a framework for determining the relative energetic contributions of the components of a regulated network. We use FP dynamics because it allows us to apply fluorescence microscopy data from living individual cells and cell populations to directly determine distributions and fluctuations. This approach precludes the need to infer transition probability rates that are needed for a master equation.

With this approach, we derive an experimentally accessible value for $\dot{Q}_{HK}$, the rate of heat dissipation associated with maintaining the nonequilibrium steady state, directly from the Boltzmann H function. While we have presented this method as applied to analyzing steady state distributions, this approach is also applicable to dynamic population state transitions as discussed in S1 Text.

Large magnitudes of correlated fluctuations of network components are associated with large rates of heat dissipation [54], and our theoretical treatment shows that this is especially true when these fluctuations occur in areas of steep landscape gradients. Noise and complexity are defining features of biological systems, and the Thermo-FP analysis suggests a thermodynamic basis for the relationship between noise and complexity and the stability of a regulated network. Fluctuations are required to ensure ergodicity by allowing cells to escape from deep attractor basins and maintain the stability of the entire landscape structure. We may consider that a biological system requires both stability and adaptability even though these may seem to be opposing characteristics. The Thermo-FP treatment presented here provides a thermodynamic basis for understanding why both the diffusion matrix and barrier gradients are important for maintaining the distribution of phenotypes. Deeper attractor basins associated with

non-negligible diffusion coefficients enable stability of the network landscape during nominally constant environmental conditions, and shallower attractors allow adaptation to changing conditions through transition to a new steady state. Very small diffusion coefficients could result in long-lived metastable states, analogous to glassy conditions.

The reasonable assumption that there will be an upper limit to the rate of heat dissipation in a biological system suggests that characteristics of landscapes and diffusion coefficients provide insight into numbers of network variables and the stability and composition of the network. There exists a geometric estimation of the number, $N$, of variables required for convergence in terms of the volumes of N-balls and N-ellipsoids [55]. Depending on characteristics of the system, such as roughness of the landscape, sufficient numbers of network terms can be predicted to be as small as a few, or many times larger. A reasonable estimate for the number of variables is between 8 and 10. How a sufficient number of terms for convergence to N can be determined is discussed in S1 Text. For example, very large landscape gradients that correspond to large eigenvalues would contribute strongly to the overall dissipative heat of the network, and a small number of such contributors may be sufficient to reach a thermal limit. This condition would be indicated by a landscape containing one or a small number of very deep attractor basins.

The measurement of dynamic correlations of many cellular variables over time in individual cells is in principle achievable with time-resolved fluorescence microscopy of live cells (see S1 Text for details), especially when enabled by automation and advances in handling of large image datasets [56, 57]. Although transcriptomics analysis can probe a larger number of genes compared to live cell imaging, the appropriate interpretation of the relative significance of these changes to network function and their relationships to one another can be ambiguous [21, 33]. Methods like transcriptomics analysis that rely on "snapshots" of populations at single points in time can infer temporal and treatment-dependent relationships between variables, but real time trajectories of changes in gene expression in individual cells, such as is accessible by live cell imaging, can provide unambiguous determination of correlations in stochastic fluctuations between network variables.

In Thermo-FP, stochasticity is captured in the dynamic fluctuations of the network variables. The magnitudes of the correlations in dynamic fluctuations provide a direct measure of the thermodynamics of network component interactions (since the magnitudes of the fluctuations are proportional to $k_B T$, Boltzmann's constant times temperature). Thus the magnitudes of the correlations are proportional to the relative energetic significance of their contribution to the network. The correlations in fluctuations between each pair of N variables of the network are assigned to an N x N diffusion matrix for each microstate. The rate of heat dissipation associated with each composite variable is determined by the magnitude of the fluctuations and the gradient of the landscape. A variable that is a negligible contributor to dissipative heat would be predicted to play a small role in maintaining the network, and could indicate that the variable is coincidental, but not causative, to network function. To achieve unambiguous interpretation, we show here how these putative relationships can be assessed experimentally by directly measuring trajectories in variable space and their covariance.

This analysis has potential practical implications. $\dot{Q}_{HK}$ is a quantitative measure of the rate of heat produced to maintain a regulatory network. The magnitude of $\dot{Q}_{HK}$ can be used to compare the relative thermodynamic cost of different steady-state phenotypic distributions. For example, as a metric it could provide insight into the thermodynamics of different regulatory networks, or the same network functioning in cells from different individuals. It will be a useful metric to guide cell therapy manufacturing conditions, and to guide the engineering of regulatory pathways in synthetic biology applications.

## Conclusions

Thermo-FP analysis, through rigorous connection to the Boltzmann H function and the rate of dissipation of free energy of the nonequilibrium steady state, provides a direct relationship between composite network variables and their contribution to the heat of maintaining homeostasis of the network.

The application of the Thermo-FP approach allows dynamical analysis of network interactions and cell states on a continuous landscape. A multidimensional (or multi-variable) landscape that considers the dynamics of network components can provide unique understanding of the correct interpretation of cellular phenotypic indicators in the context of other network components, stable attractor states and rates at which neighboring phenotypic states can be accessed. Furthermore, we have shown how $\dot{Q}_{HK}$, the dissipative heat required to maintain a multi-variable network can, in principle, be determined from experimentally tractable data consisting of a steady state distribution and a diffusion matrix of dynamic covariances in network variables. Each eigenvalue/eigenvector of the $N$-dimensional rotated matrix represents a unique and independent cluster of cooperative interactions of network components, and each constitutes a degree of freedom of the network.

Experimental observations of short-time changes in multiple network variables allows determination of the extent to which the time-dependent expression of variables is correlated. The observation of these correlations over time in individual living cells provides confirmation of causative relationships between variables.

This approach to analysis of the multivariable landscape of microstates provides unique insight into the components, paths, and the thermodynamic price associated with maintaining a nonequilibrium regulated network. While we have focused this analysis on the steady state distribution, this approach will also be useful for tracking how cellular populations transition from one steady state to another in response to environmental changes, by helping to identify and quantify interactions between network components during transitions.

## Supporting information

**S1 Text.** a. Fokker Planck dynamics and Thermo-FP framework. b. Experimental considerations. c. Convergence of the homeostatic heat to an upper bound: A geometric interpretation. d. The dissipative heat sustaining homeostasis of a network has a lower bound.
(DOCX)

## Author Contributions

**Conceptualization:** Joseph B. Hubbard, Michael Halter, Anne L. Plant.

**Formal analysis:** Swarnavo Sarkar.

**Methodology:** Joseph B. Hubbard.

**Writing – original draft:** Michael Halter, Anne L. Plant.

**Writing – review & editing:** Joseph B. Hubbard, Michael Halter, Swarnavo Sarkar, Anne L. Plant.

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
