## [Decision Letter · Decision Letter 0]

29 Oct 2019

PONE-D-19-26315

Properties of a Multidimensional Landscape Model for Determining Cellular Network Thermodynamics

PLOS ONE

Dear Dr. Plant,

Thank you for submitting your manuscript to PLOS ONE. The paper was sent to two reviewers, who have now commented upon its suitability for publication. Both reviewers have raised major concerns that should be addressed before the paper can be considered further. These concerns include, but are not limited to, incomplete reference to previous work, novelty in the light of that previous work, and relevance to biology. Please address all the comments (see below, notice also commented pdf from reviewer 1) in your resubmission.

We would appreciate receiving your revised manuscript by Dec 13 2019 11:59PM. To enhance the reproducibility of your results, we recommend that if applicable you deposit your laboratory protocols in protocols.io, where a protocol can be assigned its own identifier (DOI) such that it can be cited independently in the future. For instructions see: http://journals.plos.org/plosone/s/submission-guidelines#loc-laboratory-protocols

We look forward to receiving your revised manuscript.

Kind regards,

Jordi Garcia-Ojalvo

Academic Editor

PLOS ONE

Journal Requirements:

2. We note that your manuscript does not contain a labelled Methods section. Please consider if having a designated Methods section would be beneficial to the reader.

3. We note that Figures 1 and 2 in your submission contain copyrighted images. All PLOS content is published under the Creative Commons Attribution License (CC BY 4.0), which means that the manuscript, images, and Supporting Information files will be freely available online, and any third party is permitted to access, download, copy, distribute, and use these materials in any way, even commercially, with proper attribution. For more information, see our copyright guidelines: http://journals.plos.org/plosone/s/licenses-and-copyright.We require you to either (1) present written permission from the copyright holder to publish these figures specifically under the CC BY 4.0 license, or (2) remove the figures from your submission:

1.    You may seek permission from the original copyright holder of Figures 1 and 2 to publish the content specifically under the CC BY 4.0 license. We recommend that you contact the original copyright holder with the Content Permission Form (http://journals.plos.org/plosone/s/file?id=7c09/content-permission-form.pdf) and the following text:

Reviewers' comments:

Reviewer's Responses to Questions

**Comments to the Author**

1. Is the manuscript technically sound, and do the data support the conclusions?

Reviewer #1: Partly

Reviewer #2: No

2. Has the statistical analysis been performed appropriately and rigorously? 

Reviewer #1: N/A

Reviewer #2: N/A

3. Have the authors made all data underlying the findings in their manuscript fully available?

Reviewer #1: Yes

Reviewer #2: Yes

4. Is the manuscript presented in an intelligible fashion and written in standard English?

Reviewer #1: Yes

Reviewer #2: Yes

5. Review Comments to the Author

Reviewer #1: In their ms, Hubbard and co-workers use an FPE approach and thermodynamic arguments to investigate cell population dynamics and stability. Overall, the ms is written well but some parts would need more clear explanations as stated below and commented in the attached pdf.

While the ms is overall interesting, I have some general concerns with the content and presentation. In particular, their main result for the KL based definition of a relative entropy was derived before and a thermodynamic interpretation of high dimensional readouts was done by others, too. Therefore, the innovation of the ms is a bit limited.

Major points:

- The term Thermo-FP is a bit irritating. The FPE is rooted in statistical physics as an approximation of the Master Eq. and hence per se a TD description (including temperature which was not considered here explicitly). What the authors probably mean with the “Thermo” prefix is a thermodynamic interpretation of the fluctuation correlations? This should be clearly stated and explained in the main text.

- While their argumentation and interpretation is meaningful, the presentation could be improved to make it also readable for non-experts.

- More importantly, I missed some key references and discussions with published work.

o The connection to stochastic thermodynamic (sTD) is not done in the main text (besides one ref to Seifert in another context). In this respect, the statement that “For a nonequilibrium steady state system, the occupation of microstates results from irreversible processes ..” (l. 138) is not correct in sTD processes are reversible!

o The KL based definition of the relative free energy is rigorously derived by Rao et al. (https://journals.aps.org/prx/abstract/10.1103/PhysRevX.6.041064) and should be mentioned and compared here.

o The thermodynamic interpretation of fluctuations wrt to cell states and their stability was also studied by e.g. Chen (https://www.nature.com/articles/srep00342) or Huang (https://journals.plos.org/plosbiology/article?id=10.1371/journal.pbio.2000640).

- Given these published studies, the innovation of the ms is not really clear (and not clearly stated).

- For me the connection from the physical non-equilibrium state to the biological non-equilibrium state is not clearly described. While from the chemo-physical perspective “ENERGY” would correspond to ATP or similar energy substrates, the biological state is here described by the gene regulatory network state. How are these two levels linked? Is the physical heat dissipation comparable with biological ordering? In this context, Eqs. 3-6 should be explained in more detail to enable plausibility check. Why should Eq. 4 be general valid?

- Given these critical comments above, the proposed application to biological systems is interesting but without a proof of concept application to investigate the network contribution (even with public available data) and a concrete comparison with other methods including those mentioned above, the applicability of the suggested framework cannot be judged.

Besides these major points, I highlighted and commented some minor points in the attached pdf.

Based on this evaluation, I think that the ms is interesting but does not reach the level of PLoSONE in its current form.

Reviewer #2: This manuscript aims to develop a framework for the thermodynamics of a cell population at nonequilibrium steady states. The analysis referred to as Thermo-Fokker-Planck gives insight into the relative contributions of various network components to the relaxation process. The original method was developed for nonequilibrium steady states of nonliving systems. Efforts are made to apply this method to a population of living cells.

These are interesting ideas and the direction is important for today’s physical biology. However, it would be very useful to make the text easier to digest for both non-theoretical physicist and experimental biologist readers unfamiliar with the nonequilibrium thermodynamics bases of the manuscript.

(1) The way this method transfers from nonliving to living systems is unclear. What exactly are the sources and sinks of entropy and energy? In what sense and why are cell populations nonequilibrium? What does temperature mean here? Some clarifications are needed to make all this useful for the community.

(2) While the manuscript follows the spirit of theoretical papers such as Ref. 38 by Oono & Paniconi, it should also incorporate the spirit of Shin-ichi Sasa & Hal Tasaki, Journal of Statistical Physics 125(1), 2006, which should be cited. What Sasa & Tasaki exemplifies is how very simple, realistic systems such as sheared flow or thermal flow can be used to demonstrate the applicability of theory. The same should be done here for at least one or two biological systems: what plays the role of a “wall” (as in sheared flow) for cell populations?

(3) There are some statements that often fail in biological, cellular systems. For example, ergodicity (line 47) and detailed balance (lines 193-195) completely fail if protein levels affect the growth rate (and thereby the dilution rate). That is, the steady-state moments of time courses from tracking single-cell lineages over time will differ from steady-state moments over cell populations at any given time. This effect is described in PMID:22511863 and PMID:30341217, which would be worth citing and discussing. The statements about ergodicity and detailed balance should include the limitation that these are valid only if growth rates do not depend on protein levels.

(4) Line 141: “approaching its nonequilibrium steady state, entropy decreases over time” – this statement should be explained and references should be provided as it is unusual for anyone familiar with standard, classical thermodynamics.

(5) Figure 2: only the high-sorted population is shown over time. In addition, the unsorted population and the low-sorted populations should also be shown at the same time points.

(6) Boltzmann’s constant and temperature do not appear in the formulas of the paper. While it is OK to omit them, their meaning should still be clarified. The approach should be developed with k and T present and then they can be dropped once it is clear what happens with their incorporation. In fact, the temperature here is probably related to the fluctuations of molecule concentrations or cell states, meaning that the temperature may not be identical to the typical “absolute temperature” in statistical physics of nonliving systems. This should be clarified.

(7) It would be helpful if the method could be illustrated on a very simple, 1- or 2-dimensional system, such as a constitutively synthesized protein with or without self-regulation or something similar, using actual matrices, probability distributions, etc.

(8) Related to the previous comment, the heat terms may not be the usual heat measured in nonliving systems. This should be discussed and a practical interpretation for the heat terms should be provided.

(9) The reason for assuming the “upper bound” (line 291) should be clarified. “Temperature associated with heat generation” etc. is unclear because heat and temperature are unclear (see above).

(10) There are some typos throughout the text that should be corrected: “there can significant dynamics variability”,

6. PLOS authors have the option to publish the peer review history of their article (what does this mean?). If published, this will include your full peer review and any attached files.

Reviewer #1: No

Reviewer #2: No

---

## [Author Response · Author response to Decision Letter 0]

9 Jan 2020

RESPONSES TO REVIEWERS

We would like to begin by thanking the Reviewers for their insightful comments, which have helped greatly to improve the manuscript. We have made a number of changes to the manuscript, in addition to the comments and explanations provided below. 

In some cases, we have combined related Reviewer comments to provide a complete and concise response. We have also addressed the comments that Reviewer 1 provided in a pdf mark-up of the manuscript; these changes are embedded within the manuscript.

Responses to Reviewer 1

The term Thermo-FP is a bit irritating…What the authors probably mean with the “Thermo” prefix is a thermodynamic interpretation of the fluctuation correlations? This should be clearly stated and explained in the main text. 

Yes, the reviewer is correct, that was exactly what we intended. Thank you for the suggestion. We have made this clear by revising the title of the manuscript and adding additional explanation of the term Thermo-FP in the Abstract and Introduction.

The connection to stochastic thermodynamic (sTD) is not done in the main text (besides one ref to Seifert in another context). In this respect, the statement that “For a nonequilibrium steady state system, the occupation of microstates results from irreversible processes ..” (l. 138) is not correct in sTD processes are reversible! This sentence is ambiguous and has been deleted.

o The KL based definition of the relative free energy is rigorously derived by Rao et al. (https://journals.aps.org/prx/abstract/10.1103/PhysRevX.6.041064) and should be mentioned and compared here.

Our decomposition of dH/dt, the rate of free energy degradation, is conceptually identical to the decomposition performed in previous studies including Rao et al. We decompose dH/dt into adiabatic and non-adiabatic components, namely the housekeeping heat and entropy production rate due to transient dynamics. Our work builds on the rigorous KL based definition of the relative free energy derived by Rao. The work of Rao et al is now cited in Results.

o The thermodynamic interpretation of fluctuations wrt to cell states and their stability was also studied by e.g. Chen (https://www.nature.com/articles/srep00342) or Huang (https://journals.plos.org/plosbiology/article?id=10.1371/journal.pbio.2000640).

- Given these published studies, the innovation of the ms is not really clear (and not clearly stated).

We apparently have not articulated the innovative aspects of our theoretical framework. These innovative aspects are:

1) Our derivation for the housekeeping heat results in a quadratic form representation that provides insight into the driving force and flux that are the origin of the heat dissipation of the nonequilibrium steady state (NESS). In the form presented in Eq. 4 and 5 (previously Eq. 5 and 6), the driving force for the NESS has a clear interpretation, namely, the difference between the potential of the NESS and the corresponding equilibrium distribution (or detailed balance). The quadratic form also makes it intuitive how the fluctuations in network components, as reflected by the diffusion matrix, contribute to the housekeeping heat of the NESS. 

2) The quadratic form also lends itself to normal mode analysis, which allows identification of independent coordinated network components, the dominant modes of relaxation (degrees of freedom) of the network, that contribute to the housekeeping heat (see section “Normal Mode representation of the diffusion matrix”). As a result, this analysis uniquely applies the housekeeping heat to the interpretation of the organization of cellular networks. This implies that there is a minimum heat associated with correlated fluctuations because the quadratic form is bounded by eigenvalues of the diffusion tensor.

3) Fokker-Planck dynamics makes it straightforward to use experimental data quantifying the phenotype of cells and cell populations to analyze complex network analysis. Data on dynamic covariances in phenotype from quantitative live cell microscopy are becoming increasingly available and the corresponding landscape gradients can be obtained from steady-state cell population data. 

We have modified the manuscript (Introduction, Results, Discussion and Conclusions) to more clearly highlight these innovative aspects. 

We have also integrated the references brought to our attention by the reviewer in the Discussion section.

The work of Chen et al. and Huang et al. and others are in the same spirit as our theoretical approach in that they apply dynamical systems models to interpret transient behavior of cell populations. While it is not unusual to derive thermodynamic quantities from these types of models, our formulation of the housekeeping heat through experimental measurements of dynamic covariances is novel. Furthermore our approach is a method to apply the housekeeping heat to interpret the organization of cell networks. With respect to Chen et al. and Huang et al. our work differs in two important ways:

1) The theoretical approaches of Chen et al and Huang et al use the ensemble averages of cell population distributions, while our approach uses the dynamic covariances in phenotype that can be observed in individual cells, as well as the shape of the population distribution. By measuring changes in phenotype (such as expression of a fluorescent protein) in the same living cell over time, one can determine the fluctuation rate for that phenotypic variable and differences in fluctuation rates in different cells. By measuring more than one variable in the same cell, one can measure the correlations in fluctuations for those variables in individual cells, and the differences in correlation in different cells that may be expressing different amounts of one or the other of the variable. Access to dynamic covariance through quantitative live cell microscopy is becoming increasingly available and theoretical frameworks that can be used to interpret cell signaling networks are needed. 

2) We show how the shape of the distribution and the dynamic covariances are related to the housekeeping heat required to maintain the NESS (see description of innovative aspects above). Neither Chen et al. nor Huang et al. make connections to this thermodynamic quantity.

- For me the connection from the physical non-equilibrium state to the biological non-equilibrium state is not clearly described. While from the chemo-physical perspective “ENERGY” would correspond to ATP or similar energy substrates, the biological state is here described by the gene regulatory network state. How are these two levels linked? Is the physical heat dissipation comparable with biological ordering? In this context, Eqs. 3-6 should be explained in more detail to enable plausibility check. Why should Eq. 4 be general valid?

Our mathematical framework is applicable to both physical and biological nonequilibrium states. The constraints on the biological system could be considered to be equivalent to boundary conditions. 

The dissipative heat is indeed a result of biological ordering and the reduction of entropy. This concept is well developed by Prigogine (citation included in the text). The network is maintained by injection of energy sources or other biochemicals into the system to drive the reactions that maintain the organization of network components, and dissipative heat is released. The advantage of coarse graining with FP is that the molecular species that are involved in maintaining the network variables at their appropriate levels don’t have to be stipulated. In fact, the biological data that are collected (for example by fluorescence microscopy) provide measurable quantification of the response of network variables that could be the result of many upstream (unidentified) factors. 

The levels of network variables are measured directly in each individual cell over time; no assumptions of how the microstates are reached are necessary. The coordinated fluctuations between network variables provide the thermodynamic quantities associated with their interdependence as a function of microstate on the landscape. Eq. 4 and 5 (formerly 5 and 6) show how the coordinated fluctuations in network variables (Dij) contribute to the housekeeping heat required for maintaining the nonequilibrium steady state of the network.

In order to connect more seamlessly with other authors, We have removed the term Q.ENERGY and replaced it with Q.HK, the dissipative heat that remains associated with a nonequilibrium steady state after (dH(t))/dt has reached zero. We also modified Eq. 3 so that it is directly comparable to the housekeeping heat described by previous authors (Oono and Paniconi). 

- Given these critical comments above, the proposed application to biological systems is interesting but without a proof of concept application to investigate the network contribution (even with public available data) and a concrete comparison with other methods including those mentioned above, the applicability of the suggested framework cannot be judged.

In Sisan et al (PNAS 2012), we showed how experimental data can be used to compute gradients of the landscape and diffusion coefficients, and therefore, in the context of this work, how to compute housekeeping heat. This work follows on that study (which was on a 1-dimensional system) and describes how experimental data from multiple network variables can be collected and interpreted with respect to thermodynamic quantities. In order to make this connection between this work and the previous work, we have elaborated on the methods and results of Sisan et al in the Introduction. In that work, we showed that subpopulations that were each expressing different levels of the reporter relaxed to the steady state landscape with very different kinetics. Langevin/FP dynamics and the diffusion coefficient calculated from experimentally measured fluctuations in fluorescent protein expression allowed nearly perfect prediction of the relaxation kinetics in the absence of any adjustable parameters. This fairly remarkable finding encouraged us to develop the Thermo-FP framework for a multi-variable network system. 

Responses to Reviewer 2

This manuscript aims to develop a framework for the thermodynamics of a cell population at nonequilibrium steady states. The analysis referred to as Thermo-Fokker-Planck gives insight into the relative contributions of various network components to the relaxation process. The original method was developed for nonequilibrium steady states of nonliving systems. Efforts are made to apply this method to a population of living cells.

The current theoretical work is based on previous experimental work (Sisan et al, 2012 PNAS) of the cells in a living population. Thus, all our efforts are applicable to living systems. In order to make the connection between this work and previous work, we have elaborated on the methods and results of Sisan et al in the Introduction.

(1) The way this method transfers from nonliving to living systems is unclear. What exactly are the sources and sinks of entropy and energy? In what sense and why are cell populations nonequilibrium? What does temperature mean here? Some clarifications are needed to make all this useful for the community.

The Results section on The steady state landscape and the Boltzmann H function has been substantially edited to focus better on the thermodynamic aspects of the nonequilibrium steady state that is of primary importance here.

In this work, sources and sinks are the chemical and biochemical conditions that break detailed balance and are the origin of housekeeping heat. In order for cellular networks to maintaining homeostasis, sources of energy or other biochemicals are injected into the system and dissipative heat is released to the exterior of the system as a result. We can speculate about the identity of sources and sinks, but FP coarse graining allows analysis of the thermodynamic contributions of the network without knowing those details. The dynamic data that are collected from living cells over time (for example by fluorescence microscopy) provide a quantification of the response of multiple network variables simultaneously; these responses could be the result of many upstream (unidentified) factors. The advantage of coarse graining with FP is that the molecular species that are involved in maintaining the network variables at their appropriate levels (which are largely unknown) don’t have to be stipulated to gain predictive insight into the parameters of the network. 

We have added a citation in the introduction to support the statement: Living cells are a clear example of a nonequilibrium system (Wang 2015 Adv Phys). This citation provides many examples of nonequilibrium thermodynamics studies of cells and cell populations. 

In addition, we now make clear that by temperature we mean a real thermodynamic temperature; T is now explicitly written into the equations throughout the manuscript. We assume our systems are isothermal and that no entropy production is associated with temperature gradients. We have included this in the paper in Results.

(2) While the manuscript follows the spirit of theoretical papers such as Ref. 38 by Oono & Paniconi, it should also incorporate the spirit of Shin-ichi Sasa & Hal Tasaki, Journal of Statistical Physics 125(1), 2006, which should be cited. What Sasa & Tasaki exemplifies is how very simple, realistic systems such as sheared flow or thermal flow can be used to demonstrate the applicability of theory. The same should be done here for at least one or two biological systems: what plays the role of a “wall” (as in sheared flow) for cell populations

The concentration of biomolecules that break detailed balance (i.e. the energy sources and starting materials) are analogous to the “wall” in a sheared flow system. The constrained concentrations of biochemical species constitute the sources and sinks in our system.

Regarding an example for demonstration, we refer the Reviewer to Sisan et al (PNAS 2012). In that work, we showed how experimental data can be used to compute gradients of the landscape and diffusion coefficients, and therefore, in the context of this work, how to compute housekeeping heat. This work follows on that study (which was on a 1-dimensional system) and describes how experimental data from multiple network variables can be collected and interpreted with respect to thermodynamic quantities. In order to make this connection between this work and the previous work, we have elaborated on the methods and results of Sisan et al in the Introduction.

(3) There are some statements that often fail in biological, cellular systems. For example, ergodicity (line 47) and detailed balance (lines 193-195) completely fail if protein levels affect the growth rate (and thereby the dilution rate). That is, the steady-state moments of time courses from tracking single-cell lineages over time will differ from steady-state moments over cell populations at any given time. This effect is described in PMID:22511863 and PMID:30341217, which would be worth citing and discussing. The statements about ergodicity and detailed balance should include the limitation that these are valid only if growth rates do not depend on protein levels.

We agree with the reviewer that this is an important point regarding the application of some statistical physics concepts (i.e. ergodicity or detailed balance) to cell populations. Therefore, we have added the following paragraph to the Supporting Information 1B :

“Individual cells in a steady-state population are simultaneously transitioning between microstates and dividing. When the division rate depends on the microstate of the cell (e.g. certain phenotypes have different division rates than others), the moments of the steady-state distribution, W(x), will differ from the moments derived by following a number of individual cells over time (Ref PMID:22511863 and PMID:30341217). A microstate-dependent division rate also means that the landscape derived from the observed steady-state distribution, ln (W(x)), includes the effect of microstate-dependent division rates. An approach for compensating the landscape shape to account for microstate dependent division rates for modeling purposes was shown by Sisan et al (ref Sisan, PNAS, 2012). The landscape transformation was possible because the specific form for the dependence of the division rate on the microstate was identified by quantitative live cell microscopy.“

(4) Line 141: “approaching its nonequilibrium steady state, entropy decreases over time” – this statement should be explained and references should be provided as it is unusual for anyone familiar with standard, classical thermodynamics.

Thank you for catching this typographical error. This sentence now reads “approaching its nonequilibrium steady state, entropy increases over time”. 

(5) Figure 2: only the high-sorted population is shown over time. In addition, the unsorted population and the low-sorted populations should also be shown at the same time points.

(7) It would be helpful if the method could be illustrated on a very simple, 1- or 2-dimensional system, such as a constitutively synthesized protein with or without self-regulation or something similar, using actual matrices, probability distributions, etc.

Our previously published experimental work (Sisan et al PNAS 2012) which inspired this current theoretical work is now discussed in greater detail in the Introduction. The full data set with unsorted and the low-sorted populations is presented in that paper which is referenced. That work provides the best example of this approach. The experimental data was collected with fluorescence microscopy and flow cytometry, and the time-dependent analysis on live cells provided the data from which fluctuation rates were calculated. The paper demonstrates how the data provided both probability distributions and diffusion coefficients. These parameters alone provided a remarkably accurate prediction of the rate at which subpopulations of cells relaxed to the steady state. That study was of a one-variable system, and this theoretical work demonstrates how to build on that with a multidimensional network of associated variables. 

(6) Boltzmann’s constant and temperature do not appear in the formulas of the paper. While it is OK to omit them, their meaning should still be clarified. The approach should be developed with k and T present and then they can be dropped once it is clear what happens with their incorporation. In fact, the temperature here is probably related to the fluctuations of molecule concentrations or cell states, meaning that the temperature may not be identical to the typical “absolute temperature” in statistical physics of nonliving systems. This should be clarified.

(8) Related to the previous comment, the heat terms may not be the usual heat measured in nonliving systems. This should be discussed and a practical interpretation for the heat terms should be provided.

Temperature and kB are now explicitly written into the equations throughout the manuscript. We assume that our systems are isothermal with a reservoir and that no entropy production is associated with temperature gradients. We have included this in the paper after Eq. 1. The heat referred to is the usual heat measured in nonliving systems. The expectation is that this will be small and difficult to measure explicitly but is identical to the thermodynamic quantity. 

(9) The reason for assuming the “upper bound” (line 291) should be clarified. “Temperature associated with heat generation” etc. is unclear because heat and temperature are unclear (see above).

From a practical point of view it is reasonable to assume that there will be a limit to the rate of heat dissipation. While no physical system can withstand an infinite thermal gradient, it is well established that biological systems can be strongly influenced by temperature (Charlebois et al PNAS 2018). This citation has been added in Results.

---

## [Decision Letter · Decision Letter 1]

29 Jan 2020

PONE-D-19-26315R1

The Role of Fluctuations in Determining Cellular Network Thermodynamics

PLOS ONE

Dear Dr. Plant,

Thank you for submitting your manuscript to PLOS ONE. As you'll see from the report below, reviewer 1 still has a minor issue with the terminology that I would ask you to address before final acceptance. Upon resubmission I will send the manuscript again to reviewer 1 for a quick final decision.

We would appreciate receiving your revised manuscript by Mar 14 2020 11:59PM. To enhance the reproducibility of your results, we recommend that if applicable you deposit your laboratory protocols in protocols.io, where a protocol can be assigned its own identifier (DOI) such that it can be cited independently in the future. For instructions see: http://journals.plos.org/plosone/s/submission-guidelines#loc-laboratory-protocols

We look forward to receiving your revised manuscript.

Kind regards,

Jordi Garcia-Ojalvo

Academic Editor

PLOS ONE

Reviewers' comments:

Reviewer's Responses to Questions

**Comments to the Author**

1. If the authors have adequately addressed your comments raised in a previous round of review and you feel that this manuscript is now acceptable for publication, you may indicate that here to bypass the “Comments to the Author” section, enter your conflict of interest statement in the “Confidential to Editor” section, and submit your "Accept" recommendation.

Reviewer #1: (No Response)

Reviewer #2: All comments have been addressed

2. Is the manuscript technically sound, and do the data support the conclusions?

Reviewer #1: Yes

Reviewer #2: Yes

3. Has the statistical analysis been performed appropriately and rigorously? 

Reviewer #1: Yes

Reviewer #2: N/A

4. Have the authors made all data underlying the findings in their manuscript fully available?

Reviewer #1: Yes

Reviewer #2: Yes

5. Is the manuscript presented in an intelligible fashion and written in standard English?

Reviewer #1: Yes

Reviewer #2: Yes

6. Review Comments to the Author

Reviewer #1: While the authors have addressed most of my comments in a satisfactory manner and the ms has improved, I still do not agree with their general statement to reviewer 2 that "In addition, we now make clear that by temperature we mean a real thermodynamic temperature; T is now explicitly written into the equations throughout the manuscript." In best case, their temperature is a relative temperature that can be compared between the different system states but cannot be an absolute temperature (measured in K) because the underlying molecular mechanisms are not resolved. Hence, investigating 2 different systems or 2 different measurement wrt monitored proteins by microscopy of the same system will/can lead to different "temperatures" that can only barely be compared. This should be clarified in the text to avoid confusion of the readership. Otherwise, the ms is in a solid shape now.

Reviewer #2: I would like to thank the Authors for addressing my questions. I would like to recommend publication.

7. PLOS authors have the option to publish the peer review history of their article (what does this mean?). If published, this will include your full peer review and any attached files.

Reviewer #1: No

Reviewer #2: No

---

## [Author Response · Author response to Decision Letter 1]

30 Jan 2020

We have made an additional edit to the manuscript to address the concern of Reviewer 2. We have made a modification to our definition of T in Eq 1 and cited this use by Jarzynski in his seminal 1997 work. 

“…and T is the thermodynamic temperature of the network system in contact with an isothermal heat reservoir (40). We also assume that no entropy production is associated with temperature gradients. “

We trust that making it clear that the network is in contact with a thermal reservoir at equilibrium is the point the reviewer wanted us to make.

---

## [Decision Letter · Decision Letter 2]

21 Feb 2020

The Role of Fluctuations in Determining Cellular Network Thermodynamics

PONE-D-19-26315R2

Dear Dr. Plant,

We are pleased to inform you that your manuscript has been judged scientifically suitable for publication and will be formally accepted for publication once it complies with all outstanding technical requirements.

With kind regards,

Jordi Garcia-Ojalvo

Academic Editor

PLOS ONE

Additional Editor Comments (optional):

Reviewers' comments:

Reviewer's Responses to Questions

**Comments to the Author**

1. If the authors have adequately addressed your comments raised in a previous round of review and you feel that this manuscript is now acceptable for publication, you may indicate that here to bypass the “Comments to the Author” section, enter your conflict of interest statement in the “Confidential to Editor” section, and submit your "Accept" recommendation.

Reviewer #1: All comments have been addressed

2. Is the manuscript technically sound, and do the data support the conclusions?

Reviewer #1: (No Response)

3. Has the statistical analysis been performed appropriately and rigorously? 

Reviewer #1: (No Response)

4. Have the authors made all data underlying the findings in their manuscript fully available?

Reviewer #1: (No Response)

5. Is the manuscript presented in an intelligible fashion and written in standard English?

Reviewer #1: (No Response)

6. Review Comments to the Author

Reviewer #1: (No Response)

7. PLOS authors have the option to publish the peer review history of their article (what does this mean?). If published, this will include your full peer review and any attached files.

Reviewer #1: No

---

## [Editor Report · Acceptance letter]

28 Feb 2020

PONE-D-19-26315R2 

The Role of Fluctuations in Determining Cellular Network Thermodynamics 

Dear Dr. Plant:

I am pleased to inform you that your manuscript has been deemed suitable for publication in PLOS ONE. Congratulations! Your manuscript is now with our production department. 

With kind regards,

on behalf of

Dr. Jordi Garcia-Ojalvo 

Academic Editor

PLOS ONE